# Imputation of Missing Values for Multi-Biospecimen Metabolomics Studies: Bias and Effects on Statistical Validity

**DOI:** 10.3390/metabo12070671

**Published:** 2022-07-21

**Authors:** Machelle D. Wilson, Matthew D. Ponzini, Sandra L. Taylor, Kyoungmi Kim

**Affiliations:** 1Department of Public Health Sciences, University of California, Davis, Sacramento, CA 95817, USA; mdponzini@ucdavis.edu (M.D.P.); sltaylor@ucdavis.edu (S.L.T.); 2Department of Public Health Sciences, University of California, Davis, Davis, CA 95616, USA; kmkim@ucdavis.edu

**Keywords:** multi-biospecimen, multivariate analysis, metabolomics, mass spectrometry, missing data, imputation

## Abstract

The analysis of high-throughput metabolomics mass spectrometry data across multiple biological sample types (biospecimens) poses challenges due to missing data. During differential abundance analysis, dropping samples with missing values can lead to severe loss of data as well as biased results in group comparisons and effect size estimates. However, the imputation of missing data (the process of replacing missing data with estimated values such as a mean) may compromise the inherent intra-subject correlation of a metabolite across multiple biospecimens from the same subject, which in turn may compromise the efficacy of the statistical analysis of differential metabolites in biomarker discovery. We investigated imputation strategies when considering multiple biospecimens from the same subject. We compared a novel, but simple, approach that consists of combining the two biospecimen data matrices (rows and columns of subjects and metabolites) and imputes the two biospecimen data matrices together to an approach that imputes each biospecimen data matrix separately. We then compared the bias in the estimation of the intra-subject multi-specimen correlation and its effects on the validity of statistical significance tests between two approaches. The combined approach to multi-biospecimen studies has not been evaluated previously even though it is intuitive and easy to implement. We examine these two approaches for five imputation methods: random forest, k nearest neighbor, expectation-maximization with bootstrap, quantile regression, and half the minimum observed value. Combining the biospecimen data matrices for imputation did not greatly increase efficacy in conserving the correlation structure or improving accuracy in the statistical conclusions for most of the methods examined. Random forest tended to outperform the other methods in all performance metrics, except specificity.

## 1. Introduction

The identification of biomarkers for disease [1,2,3,4] and environmental exposure [5,6,7,8,9] using high dimensional metabolomics data is a rapidly developing field showing a high degree of promise for identifying risk factors and for developing therapeutic targets and clinical diagnostic tests. Within the last decade, the application of high-throughput mass spectroscopy (MS) to analyze hundreds or thousands of compounds from multiple types of biospecimens (e.g., serum, plasma, blood, and urine) from the same subject has increased as a method for enhancing the identification of small molecule metabolic biomarkers and understanding of physiological processes and down-stream effects in organs and pathways [10,11,12,13,14].

Commonly, -omics data from multiple biospecimens from the same subject are analyzed separately and the results compared [10,13,14,15]. However, a multivariate analysis for correlated samples that incorporates the inherent intra-subject correlation structure of a metabolite with itself across the multiple biospecimens from the same subject could enhance power and increase the information available for statistical tests of differential abundance analysis [16,17]. During statistical analysis of metabolomics data, observations with missing values are commonly dropped. This can lead to severe loss of data as well as biased results in group comparisons and effect size estimates. However, the process of replacing missing data with calculated values (imputation) may obscure the nature of inherent correlation of a metabolite across correlated biospecimens from the same subject, which in turn may compromise the efficacy of the statistical analysis in identifying abundant metabolites in relation to experimental conditions. Hence, we hypothesize that imputation methods that maintain an accurate correlation structure should provide better precision of results than those that do not.

MS data usually contain a relatively high proportion of missing values attributed to various sources [18,19,20], creating challenges that need to be addressed before statistical analysis can proceed to avoid bias in estimation of effects and statistical validity [21,22]. The missing data process for MS-based -omics data tends to be predominately non-random, e.g., censoring below the detection limit, with some portion missing-at-random due to processing errors [18]. Non-random missing (NRM) data pose additional challenges because most imputation methods assume a missing-at-random (MAR) missing data mechanism, and so may not be appropriate for MS-based -omics data. For the same reason, analyses using only observed data will not only lead to drastic loss of data with reduced sample sizes but will produce systematic bias due to left truncation or censoring by mostly discarding samples with low concentrations below detection limit. These issues may be exacerbated for multivariate analyses, where a missing value of a metabolite in one biospecimen leads to deletion of its pair in the other biospecimen in analysis of complete data (only those metabolites observed in both biospecimens), and where bias in estimation of the metabolite concentrations will affect estimates of correlation of the metabolite with its biospecimen pair and with other metabolites. Hence, accurate imputation is essential in metabolomics studies where multiple biospecimens are collected and analyzed together. 

There exists very sparse examination of the effects of missing data imputation on estimating biospecimen correlation in multivariate analysis in the literature. Taylor et al. showed that many imputation methods do not preserve the correlation structure between biospecimen data matrices and that this has biasing effects on the subsequent multivariate analyses [17]. Do et al. [23] investigated patterns of missingness in MS-based metabolomics data and found that limit-of-detection (LOD) censored data accounted for most of the missing values. As part of this study, they examined the ability of each of 31 imputation methods to reconstruct biochemical pathways in metabolic networks based on samples measured on 53 different run days, and the ability of the method to increase statistical power and preserving metabolite-gene associations. These investigations evaluated the preservation of the between-metabolite correlation structure within a biospecimen. They found that the expectation/maximization with bootstrap algorithm (EMB) [24,25] performed well, but questioned its utility given the computational challenges, and that the k-nearest neighbor (kNN) [18,26] method showed robust performance and was computationally more tractable. Imputation by quantile regression (QR) shows promise for metabolomics data [27,28,29] because this method estimates the lower quantiles of the distribution and hence should be efficacious for LOD missingness. Finally, random forest [30,31,32] has shown great potential as a non-parametric method for high-dimension data sets. If a multi-biospecimen approach is desired for assessing group differences or biomarker identification, the missing data imputation method that best preserves the inter-biospecimen correlation structure within subject should produce more reliable statistical validity. 

In this study, we used two previously published real data sets that are representative of data with properties commonly seen in metabolomics studies, one with high correlation between biospecimens and one with low, to evaluate two approaches to imputation for multiple biospecimens: (1) imputing each biospecimen data matrix (rows and columns corresponding to metabolites and subjects, respectively) separately or (2) combining the biospecimen data matrices before imputation. For each approach, we consider five different imputation methods that have either shown promise in terms of accuracy of imputation for metabolomics data or are commonly used by investigators: random forest (RF), k-nearest neighbor (kNN), expectation/maximization with bootstrap algorithm (EMB), quantile regression (QR), and half-minimum (HM). 

We used two previously published real data sets: (1) a human lung cancer data set with 178 plasma and serum metabolites (hereafter referred to as GCTOF), described in [33]; and (2) a second human lung cancer study, with 351 metabolites from lung tissue and serum (hereafter referred to as HILIC), described in [34]. Metabolites with complete data were chosen from each data set (176 from GCTOF and 327 from HILIC), which we used for our simulation studies and considered as the true complete data set for performance evaluation. 

We evaluated the two approaches (combined and separate data matrices) for each of the five methods with respect to their abilities to preserve the between-biospecimen correlation structure at different levels of missingness. Further, we assessed the influence of biased between-biospecimen correlation estimates on significance of statistical tests (sensitivity, specificity, and accuracy) by comparing the results of bivariate analysis of variance (MANOVA) of group differences (cancer versus healthy controls) using the imputed data compared to the true complete data. Here sensitivity and specificity were defined using the statistical results from the complete data: that is, true positives were defined as statistically significant results (raw *p* < 0.05) in the true data that were also significant in the imputed data. Similarly, true negatives were defined as results that were not significant in the true data that were also non-significant in the imputed data.

## 2. Results

### 2.1. Data Sets

The true GCTOF data set had 41 (23%) significantly different compounds between cancer and control groups out of 176. The true HILIC data set had 69 (21%) significantly different compounds out of 327. The GCTOF data had relatively high between-biospecimen correlations, ranging between −0.2 to 0.98, with a mode at around 0.5; whereas the HILIC data set had relatively low biospecimen correlations, ranging between −0.3 and 0.3, with a mode near 0. See Figure 1. Due to the random sampling approach to simulating missingness, some metabolites in the simulated data sets will have zero or very low percentages of missingness. These metabolites are obvious in the scatterplots that follow as they appear as perfectly or nearly perfectly imputed (i.e., they lie on or very close to the diagonal lines in Figure 2a,b).

### 2.2. Comparison of the True vs. Imputed Between-Biospecimen Correlations 

We examined the validity of each imputation method and approach by assessing the degree of concordance between the imputed and true intra-biospecimen correlations. Specifically, we estimated the between-biospecimen correlation for each metabolite for the true and imputed data.

#### Combined Versus Separate Imputation Approaches

We examined how the between-biospecimen correlation of the imputed data is similar to that of the true data, at 5%, 20%, 30%, 40%, and 60% missingness. There were no obvious differences in the results between the separate versus combined data matrix imputation approaches in either data set for any of the five methods. Regardless of the imputation method used, both approaches showed a trend of the imputed between-biospecimen correlation estimates approaching zero as percent missingness increased. Figure 2a,b show scatterplots of the between-biospecimen correlations for individual metabolites estimated from imputed and complete data, with an overlay of the two approaches. 

### 2.3. Performance of the Imputation Methods 

#### 2.3.1. GCTOF Data Set

All imputation methods performed well between 1% and 10% missingness, with no flattening of the scatter plots of between-biospecimen correlations of imputed data by those of the true data; and minimally increasing spread in the scatter with increasing missingness for both the combined and separate imputation approaches (see Figure 2a and Appendix A). At 30% missingness, the methods begin to be distinguishable with most having rapidly increasing spread with increasing missingness, mostly in a downward direction, with the imputed between-biospecimen correlations estimates decreasing towards zero for those metabolites with larger number of missing. This trend was somewhat slower for EMB, kNN and RF, whose spreads were still mostly random (rather than downward trending), with EMB having the largest spread of the three. By 40% missingness, all methods show a clear trend of estimated correlations tending towards zero for the imputed data as percent missingness increases, with RF showing the least amount of bias. 

#### 2.3.2. HILIC Data Set

Because this data set had relatively low correlations ranging from −0.3 to 0.3 and estimated between-biospecimen correlations after imputation trend towards zero with increasing missingness, the bias in this data set was lower than in the GCTOF data set at higher levels of missingness, for both approaches and all methods (see Figure 2a,b and Figure 3). However, the trends were still mostly as described for GCTOF, with increasing spread and absolute bias with increasing percent missingness, with RF performing the best as missingness increased. See Figure 2a,b and Figure 3. These results are summarized in Appendix A.

### 2.4. Bias in the Between-Biospecimen Correlation

We also assessed the efficacy in maintaining the between-biospecimen correlation structure by plotting the bias (i.e., difference of between-biospecimen correlations between the true complete and imputed data) by approach and method, and percent missingness (Figure 3). Consistent with what we described above, absolute bias increases with increasing missingness more strikingly for the more highly correlated GCTOF data set than for the lower correlations seen in the HILIC data set, due to the correlations already being near zero in the HILIC data set. The increase in absolute bias with missingness is noticeably lower for RF than for the other methods for the GCTOF data set, with EMB coming in second, for both the separate and combined approaches. 

### 2.5. Effects on Statistical Significance Tests

We assessed the effects of missingness and the resulting bias in between-biospecimen correlation estimates on significance of statistical tests (sensitivity, specificity, and accuracy) by conducting multivariate analysis of variance (MANOVA) for two-group comparisons (cancer versus healthy controls) using the imputed and true complete data. Here, sensitivity (true positive rate) and specificity (true negative rate) are defined using the significant results from the complete data: that is, true positives are defined as statistically significant results (raw *p* < 0.05) in the complete data that are also significant in the imputed data. Similarly, true negatives are defined as results that are not significant in the complete data that were also non-significant in the imputed data. 

#### 2.5.1. Sensitivity 

The performance of all methods and approaches degraded with increased missingness. Figure 4 and Figure 5 show boxplots of sensitivity and specificity for each method at each level of missingness. See Appendix A for means and standard deviations.

* HILIC data set: There were no marked advantages or disadvantages to the combined matrix imputation approach over the separate matrix imputation approach for most of the methods, with results for sensitivity varying unremarkably between the two approaches for most imputation methods, though the combined approach had slightly lower sensitivities. However, random forest performs notably better for the low correlation HILIC data set in the separate imputation approach, maintaining sensitivity of 75% even at 60% missingness. For all other methods, performance decreases rapidly with increasing missingness in both the separate and combined approaches. 

For both approaches, all imputation methods performed reasonably and approximately equally well at 1% through 10% missingness, except for QR, which consistently performed the worst of the methods examined here. The methods become more distinguishable at 20% missingness where QR and HM perform quite poorly, RF consistently out-performs the other methods, and kNN and EMB perform similarly in between. 

* GCTOF data set: For this high correlation data set, we do not see the markedly superior performance of RF with increasing missingness for the separate imputation approach, though it does out-perform the others overall. Other than for random forest, there were no notable differences between the separate and combined approaches for this data set. 

#### 2.5.2. Specificity

* HILIC data set: As with sensitivity, all methods performed well and similarly for specificity at 1% to 10%. However, contrary to the results for sensitivity, HM and QR out-perform the other methods at all levels of missingness (Figure 5). With increasing missingness, however, for this data set, one interesting difference between the separate and combined imputation approaches was that combining the matrices before imputation improved the specificity of RF, making it comparable to QR and HM. Additionally, the variability in the RF results is greatly reduced in the combined approach. Otherwise, HM and QR have superior performance in terms of specificity across all levels of missingness compared to EMB and RF (in the separate imputation), with kNN performing in the middle of the pack. 

* GCTOF: There were no obvious differences between separate and combined matrix imputation, except for improvements in specificity and the reduced variability for RF in the combined approach, similarly as seen for the HILIC data set, though not as markedly.

As with sensitivity, all methods performed well and similarly at 1% to 10%. However, as with the HILIC data and contrary to the results for sensitivity, HM and QR out-perform the other methods at all levels of missingness, with kNN being slightly lower and RF being similarly poor compared to MCM for the separate imputation and in the middle of the pack for the combined. 

#### 2.5.3. Accuracy

* HILIC: Overall RF outperformed the other methods in both the separate and combined approaches for accuracy, with the remaining methods clustering together below it. The performance of RF was improved somewhat in the combined approach, remaining above 80% accuracy even at 60% missingness, with reduced variability as seen for specificity. 

* GCTOF: Similar patterns are seen for this data set as for the HILIC, with RF remaining the best performer across levels of missingness and some reduction in the variability. See Figure 6.

### 2.6. Effects of Bias in the Between-Biospecimen Correlation Estimates on Statistical Significance Tests

RF had the least amount of bias and the least variability in the estimates of the between-biospecimen correlation after imputation. For the separate matrix approach, RF also had the highest sensitivity and overall accuracy. However, RF had the worst specificity using the separate approach, a result that was ameliorated when using the combined approach. QR and HM had similarly high bias and variability in the correlation estimates and had the lowest sensitivity and accuracy for both the separate and combined approaches. The other methods lay somewhere between these two methods. Those methods that performed well in terms of sensitivity tended to perform poorly in terms of specificity, except for RF where combining the matrices improved specificity. 

Both approaches and all five methods performed similarly across the two data sets in terms of statistical significance tests of differential analysis. This result could be because while the degree of absolute bias in the correlation estimate in the imputed data set is higher for highly correlated data, the direction of the bias is the same for both highly and lowly correlated data—that is, the imputed data tend to produce correlation estimates that are lower than the true correlation no matter which approach or method is being used.

## 3. Discussion

While little exists in the literature that addresses the issues of correlation structure in multi-biospecimen metabolomics studies, here we corroborated and expanded on results in Taylor et al. for separate imputation and further showed that combining data matrices for imputation did not result in a marked improvement in the maintenance of the inter-biospecimen correlation. We investigated the performance of multivariate analysis of metabolomics data with missing values and the impact of imputation bias and effects on statistical tests. Taylor et al. [16] found that conducting multivariate analysis on metabolites from multiple biospecimens achieved greater detection of differentially-regulated compounds and that separate imputation substantially affected the within-subject correlation of compounds across biospecimens, leading to an increased rate of false positives [17]. We investigated the simple approach of combining the multiple biospecimen matrices prior to imputation to see if this approach could better maintain the between-biospecimen correlation structure after imputation; and whether improvement in performance of significance tests—sensitivity, specificity, and accuracy—could be achieved. We additionally examined the relatively new imputation method based on quantile regression that has the potential to reduce bias in the imputation by specifically accounting for the non-random aspects of missing data from mass spectrometry due to below-detection-limit censoring. 

In general, we did not find a large degree of improvement with the combined matrix approach in terms of any of the performance metrics evaluated here, for most methods examined, nor did we find that QR had superior performance. However, some improvement in specificity, accuracy, and reducing variability was seen for RF when using the combined approach. Random forest outperformed the other methods in all metrics except specificity, with some degree of improvement with the combined approach for both data sets. Other than for specificity, QR and HM did not perform well, except at percentages of missingness 10% or lower. While it is surprising that QR did not out-perform HM, both HM and QR are aimed at accounting for lower limit censoring in the missing data process. As our simulation of missing data increases the percent missingness, we may be increasingly simulating other random processes, and this may contribute to the poor performance of HM and QR at higher levels of missingness. EMB performed poorly, while kNN performed somewhat in the middle of the pack overall. 

For very low levels of missingness (10% or less), and where the missing data process is thought to be predominantly below detection limit censoring, we recommend using HM as this method performs well and is easy to implement. For levels of missingness greater than about 10%, we recommend using RF with the separate imputation approach if the goal focuses on higher sensitivity. For studies desiring higher specificity or accuracy, such as prescreening metabolites for downstream analyses, we recommend using RF with the combined approach.

We also found that the bias in the between-biospecimen correlation estimate was determined to some extent by the degree of true correlation in the data set. Simply put, if the degree of correlation is already low, the loss of correlation that results from imputation does not greatly affect estimation. However, in the case where true correlation is high, the greater the level of missingness the more striking is the loss of the correlation structure. Random forest had the least amount of bias in its correlation estimates, the least increase in absolute bias as missingness increased, tended to have the lowest variability across simulations, and achieved the highest or comparably high sensitivity, specificity, and accuracy. 

Further research and development of methods to improve the efficacy of maintaining the intra-subject correlation structure of correlated samples are needed, as investigators increasingly use multiple biospecimens for biomarker identification and other applications to understand the complexities of disease and individual variability and to identify less-invasively or non-invasively collected biospecimens for clinical applications. 

## 4. Materials and Methods

### 4.1. Data Sets

#### 4.1.1. GCTOF

This metabolomics data set was originally generated for diagnostic biomarker discovery in lung cancer. Plasma and serum samples were obtained from 48 lung cancer and 31 normal (control) patients. Non-targeted metabolomic analyses were conducted to identify and quantify 178 metabolites. For this present study, we selected 176 metabolites with complete data (no missing values). For a full description, see Fahrmann et al. [33].

#### 4.1.2. HILIC

This metabolomics data set was obtained from lung tissue and serum samples from human volunteers with either lung cancer (38 patients) or benign nodules (40 patients). The investigators obtained samples from lung and blood. Mass spectrometry analysis and data acquisition was performed to identify and quantify 351 metabolites found in both biospecimens. We selected 327 metabolites with complete data. For a full description, see Fahrmann et al. [34].

### 4.2. Simulating Missingness 

To simulate a missing data pattern that incorporates both missing-at-random and non-random (below detection limit) missingness, we induced random missingness among lower quantiles at higher probability than for higher quantiles, for each metabolite, and holding the total missingness at the desired percent. We used restricted random sampling in an approach used by Taylor et al. [35], and similar to that described in Scheel [36] to mimic the missingness pattern of real data. To generate *p*% missingness, we randomly selected values below the *q*th quantile such that *p*% of the values in the entire data set were missing. For 1%, 5%, 10%, 20%, 30%, 40%, 50%, and 60% missing, the *q*th quantiles used were the 2nd, 10th, 20th, 40th, 50th, 60th, 70th, and 80th resulting in more missing values at low concentrations consistent with the existence of a detection limit but with no strict threshold, allowing higher concentrations to additionally be missing at random, representing random technical errors. Data sets created in this manner approximated real data sets, with some metabolites having no missing values and other metabolites with a range of missingness. Using each of the 4 complete data sets (2 experimental data sets each with 2 biospecimens), we simulated the above missing data process to create 100 data sets for each of the 4 real data sets at each of several levels of missingness. Each of the simulated data sets was then log-transformed and subjected to the imputation methods using both separate and combined approaches, and the true complete data compared to the imputed data. 

### 4.3. Metrics

To assess the effects of the imputation on statistical significance tests, MANOVA was conducted on the complete and imputed data. The complete data were used to establish the “true” significantly differentially abundant compounds. The results from the complete data and imputed data were used to calculate the sensitivity, specificity, and accuracy.

#### 4.3.1. Sensitivity (True Positive Rate)

Sensitivity was calculated as the proportion of compounds correctly identified as significantly different between the groups following imputation among those that were significantly different in the complete data.

#### 4.3.2. Specificity (True Negative Rate)

Specificity was calculated as the proportion of compounds correctly identified as not significantly different between the groups following imputation among those that were not significantly different in the complete data.

#### 4.3.3. Accuracy (True Discovery)

Accuracy was calculated as the proportion of compounds correctly identified as either significantly or not significantly different between the groups after imputation. 

### 4.4. Combined vs. Separate Imputation 

We compare two approaches (combined vs. separate) to missing data imputation for multi-biospecimen studies. The first approach is to impute missing values for each biospecimen metabolite data matrix in separate procedures for each imputation method (also known as separate imputation approach hereafter). The second approach is to combine the two biospecimen matrices and then run each imputation method on the combined matrix (also known as combined imputation approach). That is, if each separate biospecimen data matrix is *M* metabolites by *N* patients, the combined matrix is *2M* by *N*. By combining both biospecimen matrices for the imputation, the between-biospecimen correlation may be better retained. For each approach, imputation was performed using each of five different imputation methods: expectation-maximization with bootstrap, random forest, k nearest neighbor, quantile regression, and half the minimum observed value.

### 4.5. Imputation Methods

#### 4.5.1. Expectation-Maximization with Bootstrap Method

The expectation-maximization with bootstrap imputation method draws imputations of the missing values using a bootstrapping approach. The algorithm uses the familiar EM algorithm on multiple bootstrapped samples of the original incomplete data to draw values of the complete-data parameters. The algorithm then draws imputed values from each set of bootstrapped parameters, replacing the missing values with these draws [24,25,37].

#### 4.5.2. Random Forest Method

Stekhoven and Bühlmann [30] developed a missing value imputation method based on the random forest algorithm [31]. In an iterative approach, a forest is trained using observed data which is then used to predict missing values. Random forest imputation is a machine learning technique which can accommodate nonlinear associations and interactions and does not require a particular regression model to be specified. It draws on information from all observed compounds to estimate the missing values.

#### 4.5.3. K-Nearest Neighbor Method

The base assumption of kNN is that a missing value can be approximated by the values of the points that are closest to it in R^p^, where *p* is the number of metabolites observed on the neighbors. KNN imputation was initially used for microarray data [7] but has since been evaluated extensively for use in metabolomic studies [18,23,26,37]. This method finds the average of the *k* observed values for a given metabolite that are closest to the subject with the missing metabolite based on Euclidean distance in R^p^, where *p* is the number of observed metabolites. Missing values of the target compounds are imputed as the average of the *k* neighbors weighted by their distance. The optimal value for *k* can be found through cross-validation but values in the range of 5–10 are usually sufficient. By using the values of the compounds observed for the nearest neighbors closest to the individual with the missing value, kNN assesses information available across all compounds but then relies on a small set of subjects that are most like the target subject in the non-missing metabolites to impute the missing value. That is, an individual’s missing value for a given metabolite is based on the k subjects’ values that are closest to that subject in terms of the other observed metabolites.

#### 4.5.4. Quantile Regression Method

Quantile regression was recently proposed for imputing missing values in metabolomics studies [27,28]. This method assumes that compounds within a biological sample are log-normally distributed and that missing values result from detection limit censoring. Observed values are used to estimate the mean and standard deviation of an assumed log-normal distribution. Quantile regression imputes the left-censored data by randomly drawing values from a truncated normal distribution where the parameters were estimated using quantile regression.

#### 4.5.5. Half-Minimum Method

The half-minimum approach is widely used by investigators because of its simplicity [20]. The missing value of a given metabolite is imputed as half the smallest observed value for that metabolite. 

#### 4.5.6. Software

All analyses were performed using the open-source programming language, R, version 4.0.5. 

For the EMB method, we used the *Amelia* package version 1.7.6 [38] with the empirical prior setting set to the lowest value such that the algorithm converged. Following the guidance in the Amelia documentation, we set out to test empirical priors at 0.5%, 1%, 3%, 5%, 7%, and 10% of the rows to find the lowest value that converged. The HILIC data converged with the empirical prior set to 0.5% of the rows and the GCTOF data converged with the empirical prior set to 1% of the rows.

For random forest, we used the *missForest* package version 1.4 to conduct random forest imputation with default values of 10 iterations and 100 trees [30].

For K-nearest neighbors, we used 10 nearest neighbors. We set the maximum percentage of missing values for compounds at 80% above which the overall sample mean was used for imputation. We used the impute function in the R impute package version 1.64 [39].

For quantile regression, we used the *impute.QRILC* function in the R *imputeLCMD* package version 2.0 [29]. 

For half-minimum, we took half the minimum observed value for each metabolite as the imputed value for each missing value for that metabolite. 

## 5. Conclusions

There is no consistent benefit to combining the data matrices of two biospecimens from the same subject prior to imputation for multivariate analysis, though the combined approach with RF achieved superior specificity and accuracy than the other approaches. HM does well at low levels (10% or less) of missingness, so that more complicated approaches are likely not necessary for most metabolomics data with very low percent missingness. Data sets where the true correlations between biospecimens are predominantly high will suffer greater levels of bias in the estimation of the between-biospecimen correlations after imputation than data where the true correlations are lower. For levels of missingness greater than about 10%, RF with the separate imputation approach is most powerful if the goal focuses on higher sensitivity and RF with the combined approach if the goal is higher specificity or accuracy. 

## Figures and Tables

**Figure 1 metabolites-12-00671-f001:**
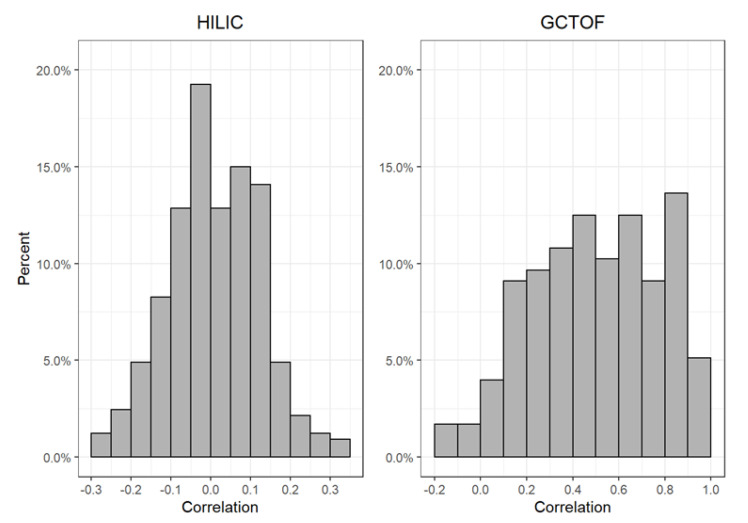
Histograms of the distribution of between-biospecimen correlations of all metabolites in the true HILIC and GCTOF data sets.

**Figure 2 metabolites-12-00671-f002:**
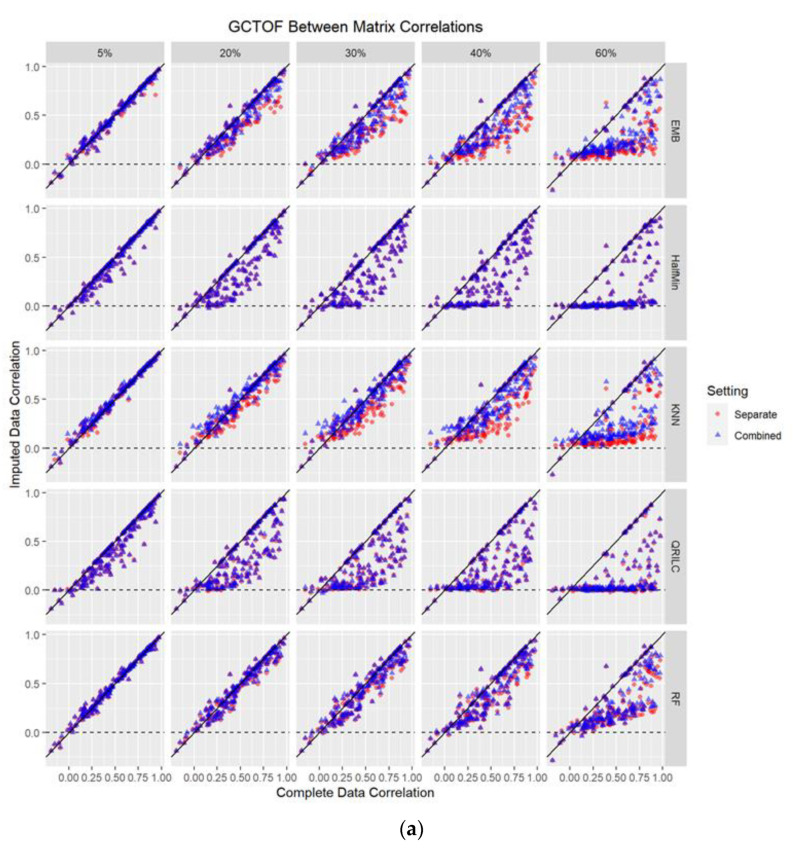
(**a**,**b**): Scatterplots of between-biospecimen correlations of the true complete data against those of the imputed data (at 5%, 20%, 30%, 40%, 60% missing values imputed) for GCTOF data set. Each dot represents an individual metabolite. EMB = Estimation/Maximization with Bootstrap imputation. Half Min = Half the Minimum observed value imputation. KNN = k Nearest Neighbors imputation. QRILC = Quantile Regression Imputation Left Censored data. RF = Random Forest imputation. (**b**) Scatterplots of between-biospecimen correlations of the true data against those of imputed data (at 5%, 20%, 30%, 40%, 60% missing values imputed) for the HILIC data set. Each dot represents an individual metabolite. EMB = Estimation/Maximization with Bootstrap imputation. Half Min = Half the Minimum observed value imputation. KNN = k Nearest Neighbors imputation. QRILC = Quantile Regression Imputation Left Censored data. RF = Random Forest imputation.

**Figure 3 metabolites-12-00671-f003:**
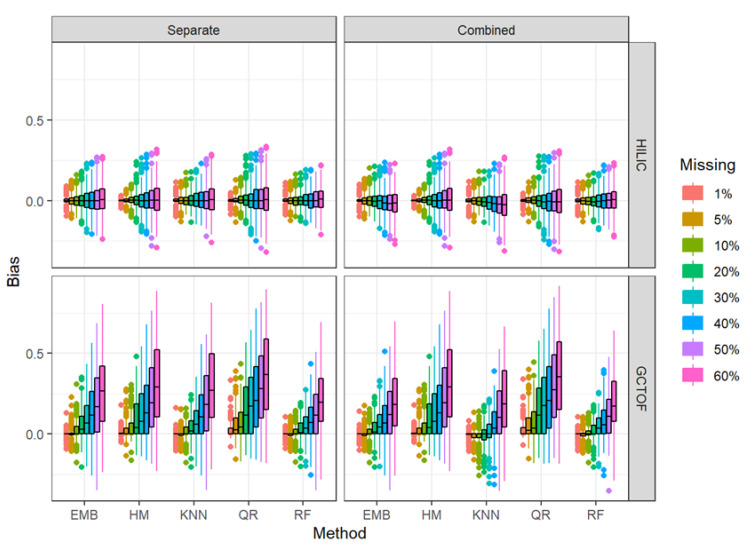
Box and whisker plots of bias in the estimated between-biospecimen correlations from imputed data for HILIC and GCTOF data sets for the separate and combined approaches.

**Figure 4 metabolites-12-00671-f004:**
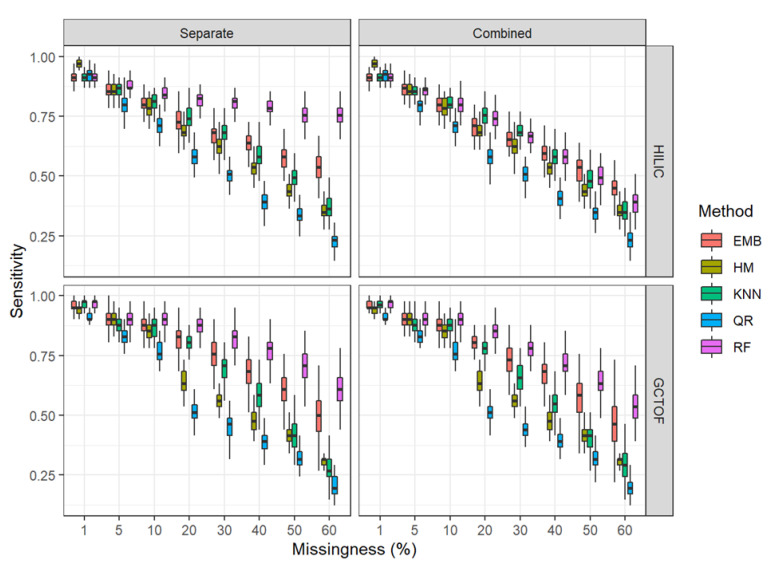
Box and whisker plots for sensitivity for the HILIC and GCTOF data sets.

**Figure 5 metabolites-12-00671-f005:**
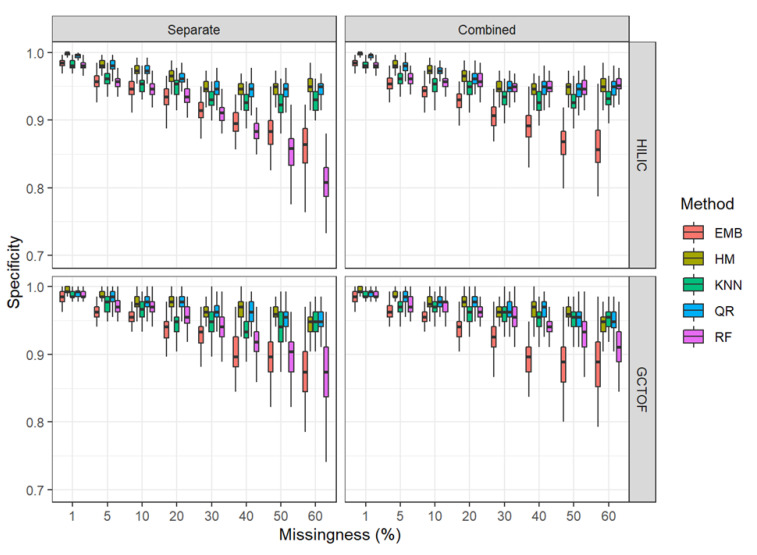
Box and whisker plots of specificity HILIC and GCTOF data sets.

**Figure 6 metabolites-12-00671-f006:**
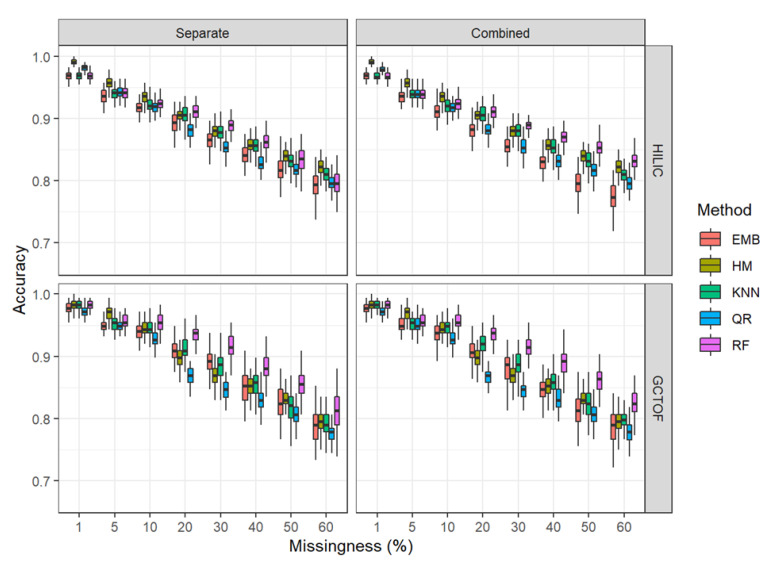
Box and whisker plots for accuracy for HILIC and GCTOF data sets.

## Data Availability

Restrictions apply to the availability of these data. Data were obtained from Suzanne Miyamoto and Karen Kelly of the UC Davis Cancer Center and are available from the authors with the permission of Suzanne Miyamoto and Karen Kelly.

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
