# Peer review of "Imputation of Missing Values for Multi-Biospecimen Metabolomics Studies: Bias and Effects on Statistical Validity"

_metabolites, 2022, doi:10.3390/metabo12070671_

Round 1
Reviewer 1 Report
This manuscript compares the effect of various imputation methods (the fact of assigning a putative value to a measurement otherwise missing for some reason) on the outcome of multivariate analysis. It compares various standard methods on 2 sets of clinic (or pre-clinic) studies. This study show some interesting results, even though the generality of these results is not verified. The study is soundly performed and seems to me of interest for Metabolites readers, and worth publishing.
The overall finding of this is that "Random Forest" is the best imputer in this kind of situation (something a lot of people had a feeling of) and that "Half Minimum" is quite efficient despite is simplistic approach. In this idea the 0 value approach (which is not so uncommon, nor so wrong for such quantitative analyses) could also have been tested.
Finally, the fact that the Quantile Regression imputation method does not show much more quality than Half-Minimum, even though it seems more adapted to the way the missing values simulation was implemented is a bit surprising and could be more discussed.
There are however problems which impose large parts of the manuscript to be rewritten.
1/ The authors seem to be more acquainted to statistical analysis than to metabolite analysis. This appears in several wording difficulties. To cite a few:
- Matrix
The word matrix is a polysemic word. It can refer to data-matrix mathematical/informatics object; but in the realm of metabolomics, it most often refers to the material from which the metabolomic analysis is performed: "material in which something is enclosed or embedded (as for protection or study)" (Merriam-Webster definition 3b) .
When I accepted the review, I genuinely thought that it was this second meaning, in particular when the abstract only refers to "multiple biological sample types", as only information on the nature of the analysis.
I recommend changing the title for something less ambiguous.
- Imputation
the word is never defined, nor even explained in a lesser sense. It appears in the introduction as if it were self-evident. I doubt it is for the mean (or median ;-) reader of "Metabolites"
- Inference, this word, which appears three times in the abstract, is used in a very technical meaning, not related the the general one (again from Merriam-Webster : "a conclusion or opinion that is formed because of known facts or evidence"), and should probably be avoided.
2/ The data-set used for the analysis is very briefly presented in the 2.1 paragraph of the Result section. It should be presented with detailed information in the M&M. (at the very least, the number of subclasses, and the number of entries in each subclasses), and maybe a classical clustering or classification analysis briefly presented, in S.I. if need be.
The last sentence of 2.1 is particularly strange, and should be developed.
3/ The Result section is otherwise not introduced, the choice to evaluate the quality of the methods on the correlation of real vs imputed values is natural, However it is not presented. All we have is a bad title: "Correlation of the Estimated Correlations", and graphs for which is is not said what is presented (what are the points ?). The the reader has to "infer" from hints in the text, what is exactly that bias presented in 2.3.2
4/ Figures 4, 5 and 6 should be redone. These figures are important, and carry a large part of the message of this work. However they are wrong (why? are the boxes shifted left-rigth relative to the % missingness (a word missing from my favourite dictionary (neologism?), but that the authors use a lot)), and misleading.
Then the evolution is not highlighted, and I had to draw lines on the top of the figures to really see what is going on).
Less important points
5/ The part of the conclusion where normative comments are given, (lines 297-303) could be highlighted.
6/ paragraph 4.3.2 should rewritten, RF imputation principle should be detailed as precisely as the other methods. The sentence "Random forest is a non-linear and non-parametric procedure that draws on information from all observed compounds to estimate the missing values." is completely uninformative and useless!
7/ is R in 4.3.3 the ensemble of real numbers ℝ ? If so, the typography should be adapted.
"p is the number of metabolites" is duplicated in this short paragraph.
Reviewer 2 Report
After carefully reviewing the manuscript, entitled “Imputation of Missing Values for Multi-matrix Metabolomics: Bias and Effects on Inference”, the authors examine these two approaches for five imputation methods: random forest, k nearest neighbor, expectation-maximization with bootstrap, quantile regression, and half the minimum observed value. Random forests tended to outperform the other methods in all performance metrics. The imputation of Missing values for Multi-matrix Metabolomics Bias and Effects on Inference are interesting. And the major comments are as followed:
1、The research novelty、significant values should been added in the abstract.
2、 In the introduction, ‘the two real data sets used were……’, the authors had not explained why GCTOF and HILIC samples were selected, and what their representativeness or reference value was.
3、About the materials and methods, it is suggested to add the calculation method of sensitivity and accuracy in the section of Materials and Methods, and attach the original data in the supplementary material.
4、Uniform format of subheadings in the whole manuscript ( eg. 4.1 and 4.3.1 are different from other subheadings).
5、In the part of discussion, it is suggested that the experimental results can be analyzed and discussed in comparing with other literatures, as well as the similarities and difference.
Round 2
Reviewer 1 Report
This version of the manuscript is a clear improvement from the initial version, in particular the introduction.
Ambiguities in the wording have been cleared, important notions such as imputation are introduced. M &M is now correctly developed and Figure captions more descriptive.
I still think that Fig 4-5-6 could have been made more informative, but the information is present nevertheless.
I consider that the work can be published in its present state
Reviewer 2 Report
The authors changed the manuscript accordingly to the previous comments and improved it significantly. So, the manuscript is suitable for publication.